# High-order coherent communications using mode-locked dark-pulse Kerr combs from microresonators

Attila Fülöp[1], Mikael Mazur[1], Abel Lorences-Riesgo [1,4], Óskar B. Helgason[1], Pei-Hsun Wang[2], Yi Xuan[2,3], Dan E. Leaird[2], Minghao Qi[2,3], Peter A. Andrekson[1], Andrew M. Weiner[2,3] & Victor Torres-Company [1]

Microresonator frequency combs harness the nonlinear Kerr effect in an integrated optical cavity to generate a multitude of phase-locked frequency lines. The line spacing can reach values in the order of 100 GHz, making it an attractive multi-wavelength light source for applications in fiber-optic communications. Depending on the dispersion of the microresonator, different physical dynamics have been observed. A recently discovered comb state corresponds to the formation of mode-locked dark pulses in a normal-dispersion microcavity. Such dark-pulse combs are particularly compelling for advanced coherent communications since they display unusually high power-conversion efficiency. Here, we report the first coherent-transmission experiments using 64-quadrature amplitude modulation encoded onto the frequency lines of a dark-pulse comb. The high conversion efficiency of the comb enables transmitted optical signal-to-noise ratios above 33 dB, while maintaining a laser pump power level compatible with state-of-the-art hybrid silicon lasers.

[1] Photonics Laboratory, Department of Microtechnology and Nanoscience, Chalmers University of Technology, SE-41296 Göteborg, Sweden. [2] School of Electrical and Computer Engineering, Purdue University, West Lafayette, IN 47907-2035, USA. [3] Birck Nanotechnology Center, Purdue University, West Lafayette, IN 47907-2035, USA. [4] Present address: IT-Instituto de Telecomunicações, 3810-193 Aveiro, Portugal. Correspondence and requests for materials should be addressed to V.T-C. (email: torresv@chalmers.se)

Replacing a large number of lasers in wavelength-division-multiplexed (WDM) optical communication systems with an optical frequency comb has always been an attractive idea. Until recently, demonstrations focused on broadened mode-locked lasers[1,2] and electro-optic frequency combs[3] made using a cascade of phase and intensity modulators[4]. Electro-optic comb generators can use a single high-quality laser as a seed and replicate its properties to several channels. Increasing the bandwidth further is possible by using nonlinear broadening[5–7], allowing for lighting more WDM channels. Optical frequency combs have an intrinsically stable frequency spacing that enables transmission-performance enhancements beyond what is possible with free-running lasers. Recent demonstrations include multi-channel nonlinear pre-compensation[8], as well as the possibility to decrease the inter-channel guard bands for an increased total spectral efficiency[9,10]. Another exciting prospect for using a frequency comb as a multi-carrier light source in WDM systems is the possibility to relax the resource requirements at the receiver by implementing joint impairment compensation and tracking for multiple data channels[11,12]. This aspect exploits the broadband phase coherence of the frequency comb and is therefore impossible to carry out using a multi-wavelength laser array.

In order to implement practical WDM transmitters while minimizing the number of discrete components, photonic integration will be needed. It is however challenging to attain broad bandwidth and high-powered chip-scale frequency combs with similar levels of performance as in the demonstrations above. Initial attempts have included silicon–organic hybrid modulators[13], quantum-dash mode-locked lasers[14], and gain-switched laser diodes[15]. A CMOS-compatible platform that shows great promise in this direction is the microresonator frequency comb implemented in silicon nitride technology[16]. Microresonator frequency combs (or Kerr combs) use the Kerr effect in an integrated microcavity to convert light from a continuous-wave pump laser to evenly spaced lines across a wide bandwidth[17–20]. The first data-transmission demonstrations were performed using on–off keying[21,22]. It was however soon recognized that the performance of microresonator combs is sufficiently high to cope with the requirements in terms of frequency stability, signal-to-noise ratio (SNR), and linewidth of modern coherent communication systems[23–26]. Recent demonstrations have therefore included advanced modulation formats[23,24] and long-haul communication systems[26]. The discovery of dissipative Kerr solitons in microresonators[27] and associated stabilization schemes[28–30] has opened a path forward to control the bandwidth and number of comb lines with great precision. One of the most recent experiments has achieved impressive aggregate data rates using two SiN microresonator combs spanning the lightwave communication of C and L bands[31]. Using thermal control, tuning of the central frequency of the combs has allowed the use of a matched comb at the receiver as a multi-wavelength local oscillator[31].

Microresonator frequency combs are however complex systems that permit several different regimes of low-noise operation. The coherent communication demonstrations thus far have mainly focused on combs operating in the bright soliton[25,28] and coherent modulation instability[26,32] regimes. Recent experiments have revealed a mode-locked state when the cavity exhibits normal dispersion. This comb state corresponds to circulating dark pulses[33,34] in the cavity, and it might be of significant interest for coherent data transmission in WDM systems. These dark-pulse combs have experimentally measured power-conversion efficiencies between the pump and the generated comb lines above the 30% mark[35] for combs spanning the C band, i.e., significantly higher than what can be fundamentally obtained with bright-soliton combs[36]. If these powers could be harnessed, such dark-pulse combs could either decrease the pump power requirements

or enable WDM transmitters with higher SNRs. This is a relevant matter as modern communication systems move toward even more advanced, higher-order modulation formats. These formats contain increasing amounts of encoded data per transmitted symbol, which results in increased requirements in the received SNR[37].

In this work, we show what we believe is the first coherent WDM transmission experiment conducted with a dark-pulse Kerr comb. We use a SiN-based microresonator that produces comb lines satisfying the optical signal-to-noise ratio (OSNR) requirements for modern coherent communication formats. The high OSNR per channel is enabled by the high internal conversion efficiency of the comb, which reaches above 20%–in line with previous observations[38]. Using off-chip pump powers below 400 mW, we demonstrate 80-km data transmission with 20 channels. Each channel contains data encoded using 20-GBd 64-quadrature amplitude modulation (QAM), resulting in an aggregate data rate of 4.4 Tb/s (assuming a 9% error-correction overhead). This demonstration corresponds to the highest-order modulation format shown with any integrated comb technology to date.

## Results

**Microresonator-based frequency comb generation**. A silicon nitride-based microresonator is used for the comb generation. The resonator's 100-μm radius results in a free spectral range of about 230 GHz. The ring waveguide features a designed width and thickness of 2 μm and 600 nm, yielding normal dispersion in the C band[32]. Fabrication details are described in ref. [39]. The intrinsic Q-factor is measured to be around 1.6 million. For the comb generation, the microresonator is pumped by a tunable external-cavity laser with less than 10 kHz of specified linewidth. Before reaching the microresonator, the laser light is amplified and filtered, ensuring an off-chip continuous-wave pump of 25.6 dBm. Figure 1a shows a sketch of the setup used for the comb generation. The fiber-to-chip coupling losses at high pumping powers are estimated to be 5 dB per facet. The microresonator is equipped with both a through and a drop port, with the latter being used for assessing the intracavity waveform. As the coupling between the resonator and the through port is stronger, the comb obtained at the through port is used for the communication experiments (see Methods).

The comb is initialized by tuning the pump laser wavelength into a resonance located at 1540 nm from the thermally stable blue side[40]. To monitor the running comb state, a photodiode is placed after an optical band-pass filter centered at a newly generated comb line around 1536 nm, see Fig. 1a. By using the photodiode output as feedback to the laser wavelength setting, it is possible to start the comb by placing the laser close to resonance and simply initializing the lock. This way, the pump will stop sweeping at the moment the comb is in the desired state. Although locking is not necessary to keep the comb running over several hours, the feedback loop ensures that laboratory environmental changes do not cause the spectrum to change significantly over this time[38]. The spectrum of the generated comb at the through port, displayed in Fig. 1b, shows the characteristic envelope of dark pulses[33,34,41]. To verify that the intracavity comb state corresponds to a circulating dark pulse, two separate time-domain measurements are taken at the device's weakly coupled drop port. A direct measurement performed using a 500-GHz bandwidth optical sampling oscilloscope results in the waveform shown in Fig. 1c, indicating square-like pulses. An effectively higher bandwidth, but indirect measurement, is also taken by measuring the comb lines' spectral phase, after which the time-domain picture is reconstructed[33,42] as shown in Fig. 1d. The Methods section contains a more detailed description

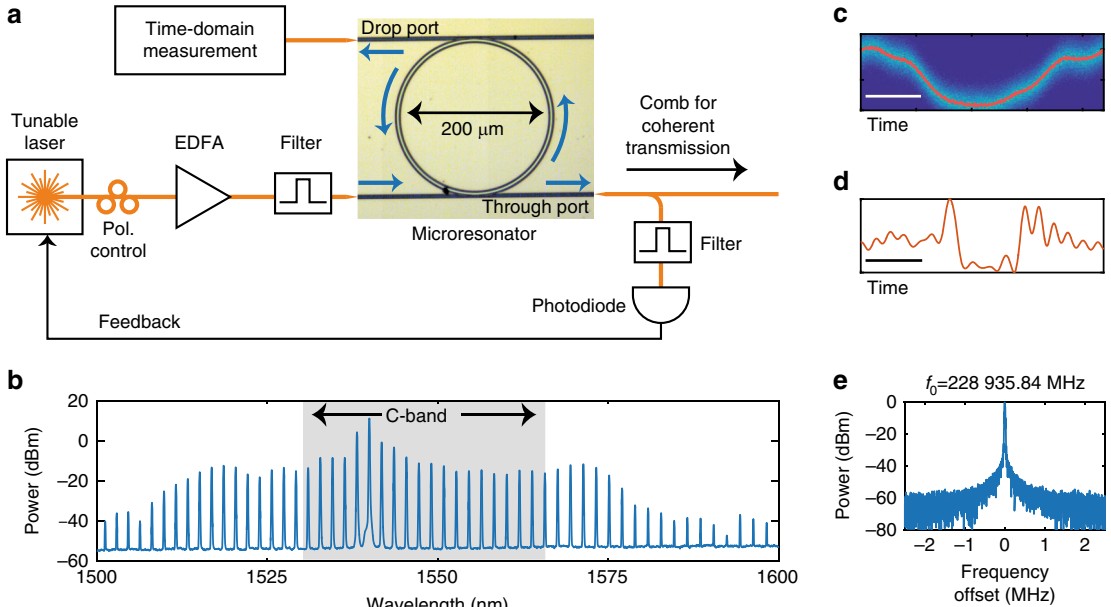

**Fig. 1** Frequency comb operation. **a** Setup for generating and measuring the comb state. The amplified spontaneous emission noise generated by the erbium-doped fiber amplifier (EDFA) is removed using a 200-GHz band-pass filter centered at the pump wavelength. The comb state is stabilized against long-term drifts by selecting a line of 460 GHz above the pump frequency using a second 200-GHz band-pass filter. **b** Comb spectrum measured at the through port with 0.1-nm resolution. The 20 comb lines in the shaded region between 1531 and 1566 nm are used for the communication experiments (the C band is typically specified between 1530 and 1565 nm). **c** Directly measured and **d** reconstructed intracavity time-domain waveforms using the drop port of the microresonator. The scale bars correspond to 1 ps. **e** The comb-line spacing measured with 1-kHz resolution

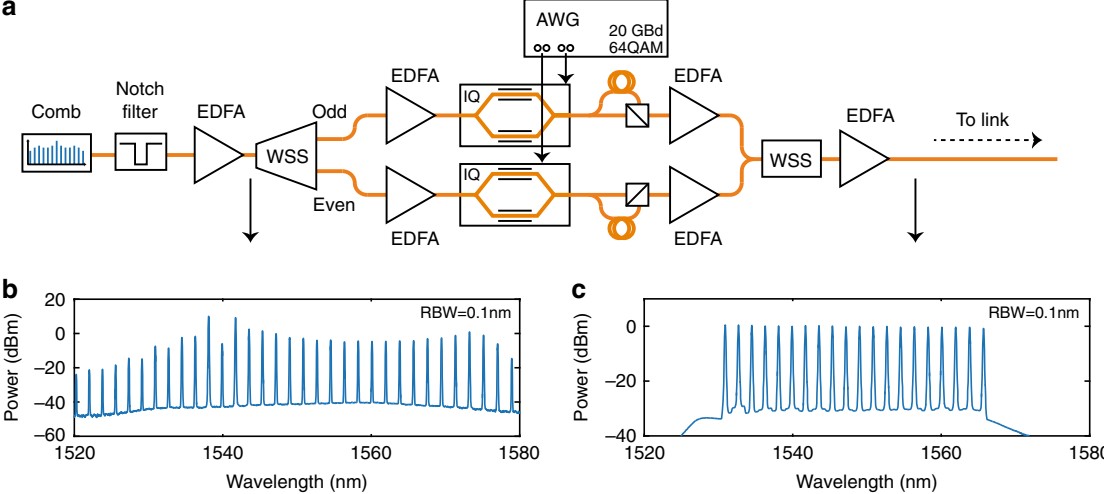

**Fig. 2** Transmitter description. **a** Schematic of the transmitter. WSS wavelength-selective switch, AWG arbitrary waveform generator. **b** Optical spectrum after pump line attenuation and the single-stage amplifier. The 20 lines of interest all have an OSNR of 35 dB or more. **c** Optical spectrum of the WDM signal before being sent to the link with all channels having above 33 dB of OSNR

of the measurement and reconstruction procedure. In addition, the comb line spacing stability was measured using electro-optic downconversion[43]. The beat note (see Fig. 1e) displays a clear peak > 50 dB above the noise floor (with a Gaussian fit FWHM < 30 kHz), indicating stable mode-locking operation beyond what is required in state-of-the-art dense WDM demonstrations[44].

**Optical data modulation**. The microresonator comb described in the previous section is now shown to support data transmission with advanced modulation formats. To ensure maximum comb-line powers for the data-transmission experiment, we used the dark-pulse comb at the through port as a light source for the

transmitter. At this port, the total fiber-coupled comb power is roughly 28 mW (out of which about 8.6 mW is in the newly generated comb lines), leading to an on-chip comb power-conversion efficiency above 20% and a flattened net conversion efficiency of 1.5%, see the Methods section for calculation details. Following the chip itself, a 200-GHz notch filter centered at the pump wavelength attenuates the central comb line, allowing for an efficient operation of the following optical amplifier. For the 80-km single-span WDM experiment, the transmitter's schematic is shown in Fig. 2a with the initially filtered and amplified comb spectrum visible in Fig. 2b. Following amplification, the power in the comb lines is split into two arms (marked as odd and even in

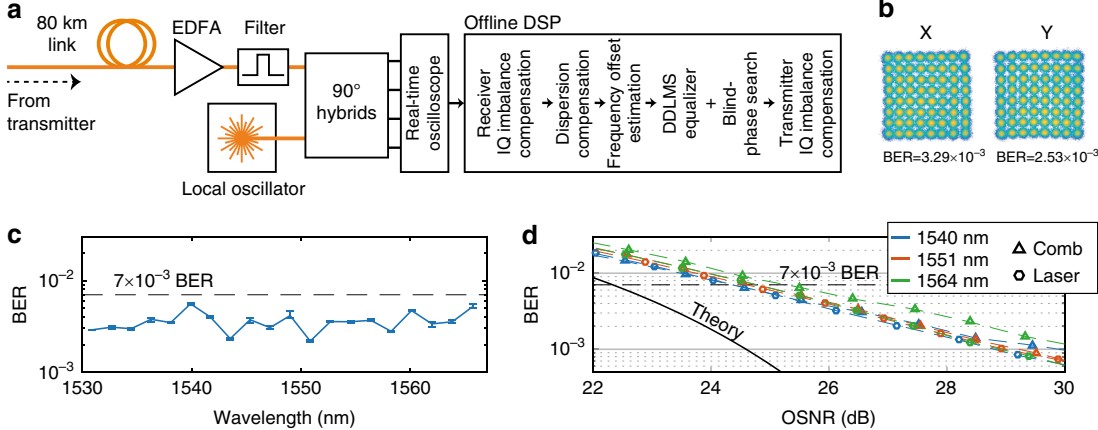

**Fig. 3** Receiver and results. **a** Schematic of the receiver including the blocks in the offline DSP. **b** Received constellations from the 1531-nm line. The constellations are displayed for both polarizations (marked X and Y) at the corresponding bit error ratios (BERs) of $3.29 \times 10^{-3}$ and $2.53 \times 10^{-3}$. **c** Received BERs averaging both polarizations for five batches for each comb line after the 80-km-long transmission link. The error bars show the BER values for the best and the worst batch for each wavelength. The wavelength-dependent BER variations are within the expected levels considering the transceiver components. **d** Noise-loading measurements for three selected comb lines and comparison measurements using a free-running laser. The black full-drawn curve corresponds to the theoretical additive white Gaussian noise channel case assuming Gray coding[61]. The dashed curve shows the maximum permitted BER, allowing for a 9.1% overhead for the implementation of a hard-decision staircase FEC code[45]. All measurements were performed with random data modulated using 20-GBd PM-64QAM

the setup), using a commercially available wavelength-selective switch (WSS). This keeps the number of lines going into each modulator at ten. The WSS is also used to equalize the powers among the comb lines in each arm separately. Each modulator is driven by signals generated in an arbitrary waveform generator (AWG). The AWG is programmed to generate two independent random 64-QAM signals using square pulses, each carrying 6 bits per symbol at a rate of 20 GBd. The random symbol sequence has a length of $2^{16}$ symbols and an oversampling of three since the AWG is operated at 60 GS/s. To mitigate non-idealities in the digital-to-analog converters (DACs) as well as the modulators, digital pre-compensation is applied on the signal in the AWG. Following the modulators, a split-and-delay polarization-multiplexing stage, using a ≥1-m long arm corresponding to ≥100 data symbols, is used to emulate a polarization-multiplexed transmitter. By using both polarizations, the system capacity is effectively doubled. Both arms are then recombined and flattened using a second WSS before being sent to the link. The second flattening step will translate the power differences in the two arms (caused by the comb envelope and the initial flattening step) into the slightly varying noise floor seen in Fig. 2c.

**Transmission results**. After the 80-km-long standard single-mode fiber link (with about 16 dB of propagation loss), there is a polarization-diverse single-channel coherent receiver, see Fig. 3a. A tunable external-cavity laser (with linewidth below 100 kHz) is used as a local oscillator, allowing for the reception of one data channel at a time, using a 23-GHz bandwidth real-time oscilloscope operating at a sampling rate of 50 GS/s. Standard digital signal-processing (DSP) algorithms are subsequently run offline and are described in more detail in the Methods section. A representative example of a received constellation is presented in Fig. 3b. To ensure optimal signal power in the fiber, data are recorded for multiple launch powers. The results in Fig. 3c correspond to the optimum case, with a launched signal power of 3 dBm per channel. Higher power levels incur nonlinear distortion, whereas lower powers lead to a system performance limited by noise. The resulting BERs are calculated by comparing the decoded bit stream with the transmitted one. All comb lines provide sufficient BER margin for transmission over an 80-km fiber. The resulting BER values allow the application of a hard-

decision staircase forward error correcting (FEC) code[45] with an overhead of 9.1% to reach a final post-FEC BER below $10^{-15}$, yielding an aggregate data rate of 4.4 Tb/s.

The final experiment corresponds to a configuration where there is no transmission fiber ("back-to-back"). This experiment serves two purposes. First, it allows quantifying the performance of the transmitter/receiver with respect to an idealized situation where only additive white Gaussian noise is considered (theory). Second, it allows for distinguishing penalties coming from the comb source and from the transceiver subsystems by comparing the results with a similar measurement performed using a stand-alone laser. The measurement is performed on a channel-by-channel basis by measuring the received BER for varying OSNR. The setup and measurement details are described in the Methods section. As shown in Fig. 3d, at the chosen BER threshold of $7 \times 10^{-3}$, the comb lines require a slightly higher received OSNR (between 0 and 0.5 dB) compared with the free-running laser system. Both the comb lines and reference lasers show that an increase in OSNR by 3 dB with respect to the theoretical prediction is required to reach the same target BER value (this is often referred to as implementation penalty). Such a deviation from the theoretical case is expected for advanced modulation formats with high symbol rates, where the limited effective number of bits in both the transmitter and the receiver electronics impairs the transmitted signal[46,47]. These results indicate that the microresonator comb source does not significantly impair the transmission link performance and is therefore a suitable light source for higher-order coherent optical communication systems.

## Discussion

In summary, we have presented the first demonstration of coherent WDM communications using dark-pulse microresonator combs. We have shown the highest-order modulation format demonstrated using any integrated comb source. An important aspect of this study is that it illustrates that the favorable power-conversion efficiency of dark-pulse combs can be used in practice to reach channel OSNRs >33 dB, while maintaining an on-chip pump power in the order of a few hundred mW.

While in this exploratory study the setup complexity was extensive and the spectral efficiency was relatively low (about

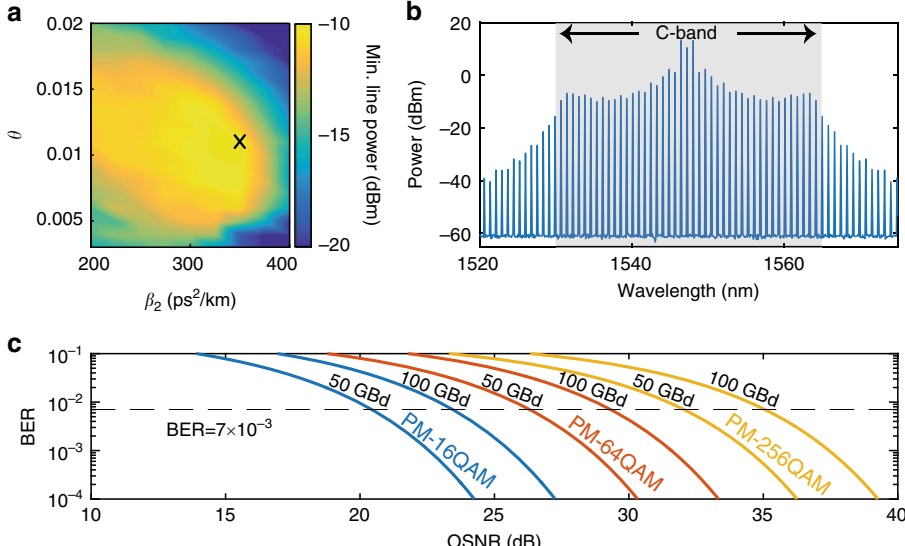

**Fig. 4** Dark-pulse comb optimization for modern communication formats. **a** Simulation results showing the minimum comb-line powers as the waveguide's group velocity dispersion coefficient, $\beta_2$, and the ring waveguide coupling constant, $\theta$, are varied. The comb spacing is fixed at 100 GHz. The peak value marked with X at −10 dBm is achieved for $\beta_2 = 350\ ps^2/km$, $\theta = 0.011$, and a detuning of $\delta_0 = 0.0445$, yielding a comb with above 50% power-conversion efficiency and a flattened net conversion efficiency of 4.4%. **b** Optical spectrum of the optimized dark-pulse comb with line powers in the C band between 51 dB and 71 dB above the quantum noise limit at 0.1-nm resolution. **c** Theoretical OSNR requirements at the receiver side for a variety of modern communication formats and symbol rates, assuming additive white Gaussian noise and Gray-level coding[61]. For a single-polarized signal, the limits will be the same assuming that the noise is only measured in that polarization. We note that a doubling in the symbol rate or modulation format results in roughly a 3-dB increase in the OSNR requirement

0.95 b/s/Hz), these are not fundamental limits of the capabilities of microresonator comb-based transmission systems. In this section, we analyze the fundamentally achievable OSNRs per wavelength channel for optimized dark-pulse combs. We demonstrate that the achievable OSNR levels are compatible with state-of-the-art hybrid silicon tunable lasers, which feature optical linewidths of 15 kHz and power levels in the order of 100 mW[48]. The resulting OSNRs are sufficiently high to encode higher-order modulation formats while leaving sufficient margin to allow for losses when the comb is co-integrated with other active components in a transmitter system[23,49].

For data-transmission purposes, the weakest line power in the comb will fundamentally dictate the minimum achievable OSNR per line at the transmitter[31]. The line's power level can be maximized by jointly optimizing the group velocity dispersion coefficient and coupling rate of the microresonator. We ran an optimization process (see Methods) of a dark-pulse comb spanning the C band by sweeping these two parameters while keeping the microresonator losses, pump power, line spacing, and nonlinear parameters fixed. The pump power was fixed to 100 mW, and a line spacing of 100 GHz was selected that is more compatible with WDM standards. Assuming that challenges involving propagation loss and multimode behavior can be handled, a larger resonator with lower FSR could in principle result in a dark-pulse comb covering the C band with a line spacing closer to 100 GHz. Recent demonstrations[50] indicate that this is indeed possible. The result of the optimization process is shown in Fig. 4a and it shows that an optimum state can be found for a moderate amount of normal dispersion and a strongly coupled microresonator. The fabrication feasibility of these parameters is discussed and verified in Supplementary Note 1. The weakest line power within the C band appears at the −10-dBm level and the resulting comb is shown in Fig. 4b. The comb lines have a power level varying between 51 and 71 dB above the quantum noise limit (corresponding to around −61 dBm for a single polarization

and 0.1-nm bandwidth). Supplementary Note 2 contains results using the same optimization process for combs with 50 GHz of line spacing, as well as combs covering both C and L bands.

Receiver OSNR requirements for advanced modulation formats and symbol rates are displayed in Fig. 4c. For example, 50-GBd PM-64QAM would require an OSNR at the receiver side of 26 dB. Assuming that the WDM transmitter is amplified with an EDFA with a noise figure of 4 dB and considering 5 dB of implementation penalty due to the limited effective number of bits at the transmitter and receiver yields a margin of 16 dB for the weakest comb line. If the same microresonator were used as a source for signals in both polarizations, a further 3 dB should be deducted. The remaining 13-dB margin is to be split between optical losses at the integrated transmitter (containing multiplexers and modulators) and the effects of the following link (owing to added noise by further amplification as well as nonlinear effects[51]). Given the high available power in the central lines, these can carry data using even higher-order modulation formats, potentially targeting PM-256QAM. A transmitter scheme where the power in the stronger lines can be exploited in this manner is discussed in Supplementary Note 3. We envision that further device optimizations (i.e., larger resonators with optimal coupling and chromatic dispersion values) together with advances in integrated photonic circuits[49], will allow approaching a more complete integrated transmitter. By additionally restricting the system requirements to around 100 mW of pump power, the co-integration of a pump laser would become feasible. We also anticipate the expected channel symbol rates to keep increasing in the near future, making a 100-GHz spaced comb as the final target. As described in this work, an optimized dark-pulse microresonator comb has the potential to empower such a system.

## Methods
**Indirect time-domain measurements**. The indirect time-domain measurement was done by sending the frequency comb through a WSS and an EDFA to a non-colinear optical intensity autocorrelator[33,42]. By selecting three lines at a time in the

WSS and adjusting their phases incrementally to achieve minimum pulse width in the autocorrelation measurement, we could extract the relative phases across all comb lines within the bandwidth of the EDFA and the pulse shaper. The measurements were performed using both the drop and the through ports of the microresonator, with the resulting phases overlapping for all measured lines except for the pump line. Using a known mode-locked laser delivering transform-limited pulses as a reference, the dispersive effects of the fibers in the EDFA, the WSS, and the fiber connections were measured and compensated for. While most of the power in the comb was within the bandwidth of the EDFA and the pulse shaper, the spectral phases of the lines outside were estimated using linear extrapolation.

**Line spacing stability measurements.** As the frequency difference between the comb lines was too large for our photodiodes' bandwidth, a direct measurement was not possible. Instead, the spectrum between the lines was filled using electro-optic (EO) modulation[43]. We selected two central comb lines with a 2-nm optical band-pass filter (at 1539.8 and 1541.6 nm) and generated EO-modulated lines between them using an RF oscillator operating at 25.1 GHz. The beat note between the two resulting EO-comb lines at 1540.8 nm (one originating from each original dark-pulse comb line) was then filtered out using a 0.25-nm filter and recorded using a real-time oscilloscope. From 1 ms of recorded data, we could thereby retrieve the spectral stability of the comb-line spacing with 1-kHz resolution by standard Fourier processing.

**Microresonator operation.** The chip containing the microresonator was kept on a piezo-controlled positioning stage stabilized using a standard laser temperature controller at 18 °C with <0.01 °C variation. Light was then coupled into the bus waveguide using a lensed fiber. On the chip, the coupling between the bus waveguide and the ring was set by their 300-nm wide gap. As the drop-port gap (at 1000 nm) was much larger, the power coupling to that port is estimated to be more than 10 times weaker (see Supplementary Note 1). The round-trip losses owing to light getting coupled out of the resonator are therefore expected to be dominated by the through port.

As the comb-generation process strongly depends on the presence of modal coupling[52], the spectral envelope is sensitive to slight fabrication variations. Out of three similar devices, two produce dark-pulse combs using a 1540-nm pump. The tolerances (and thereby the device yield) are however expected to be improved, using techniques enabling post-fabrication tuning, for example, using controllable mode interactions as described in ref. [53].

**Comb power-conversion efficiency calculation.** To estimate the power-conversion efficiency of the comb-generation process, both the comb state and the reference off-resonant state have to be compared under identical pump power and polarization conditions. The conversion efficiency is calculated by comparing the sum of the power in the generated comb lines in the on state with the power of the pump line in the off state[26].

To estimate the useful net conversion efficiency, one can perform a similar calculation. Instead of summing the power of all the newly generated comb lines, one should instead take the power in the weakest line within the bandwidth of interest (the C band in this case) and multiply that with the number of lines present within the same bandwidth (20). Comparing this power with pump line power in the off state will yield the effective flattened on-chip conversion efficiency.

**Digital signal-processing algorithms.** To decode data from the recorded complex waveforms, receiver non-idealities, link effects, and transmitter non-idealities have to be handled, in that order. The following steps describe the DSP operations:

1. Receiver impairments were handled by first compensating for relative delays owing to differences in the RF cable lengths in the I and the Q arm of the coherent receiver. This was followed by an IQ imbalance compensation using Gram–Schmidt orthogonalization[54]. After this, the waveform was resampled to twice the symbol rate. All these steps were performed independently for each polarization.
2. In the case of the 80-km-long transmission, chromatic dispersion of the single-mode fiber link was removed using a static filter implemented in the frequency domain individually for each polarization.
3. A decision-directed least-mean-square equalizer was used for signal equalization and polarization demultiplexing. This step also included an FFT-based frequency offset estimator[55] and a blind-phase search component to compensate for relative phase drifts between the signal carrier and the local oscillator[56]. The equalizer contained 25 taps and was trained by letting it run over the waveform four times with decreasing step length.
4. A second Gram–Schmidt orthogonalization was performed individually on each polarization to compensate for small modulator bias errors.
5. Finally, the BER was calculated by comparing the bit sequence decoded from the received symbols with the originally transmitted one. The total received sequence from which the BER was calculated contained more than 9 million bits.

**Noise-loading measurements.** The noise-loading measurements allow comparing the performance of single channels with theory to extract quantitative penalties

accrued owing to the system implementation (occurring for example due to the limited resolution in the transmitter digital-to-analog converters and receiver analog-to-digital converters). To isolate these penalties from those coming from the comb source, separate measurements were taken with individual comb lines as well as a reference laser. For the evaluation of our transmitter and receiver system, a single tunable external-cavity laser was therefore used. The reference laser corresponded to a standard (below 100 kHz of linewidth) communication laser with above 15 dBm output power, nominally identical to the local oscillator. The reference laser was connected directly to the modulator in one of the arms in Fig. 2a. Following the polarization-multiplexing stage, the channel was loaded with noise by successive attenuation and amplification. Finally, the channel was then received, resulting in the BER vs. OSNR plots in Fig. 3d, yielding a ≤2.5-dB system penalty with respect to theory at BER = $7 \times 10^{-3}$. To make an equivalent measurement with the comb source meant selecting single comb lines and amplifying them to the same 15-dBm power level before performing data modulation.

**Dark-pulse comb simulations.** The comb-state simulations were performed using the Ikeda map[57–59]. This method allows for including the pump noise in every round trip and to quantify the resulting OSNR per spectral line. The method involves simulating the coupling between the bus waveguide and the ring separately from the light propagation inside the ring cavity. The coupling region was simulated as a lossless directional coupler[60], whereas the propagation in the ring was implemented with the nonlinear Schrödinger equation. In the coupling step, a continuous-wave pump laser (with 100 mW of fixed power) was coupled together with quantum noise equivalent to one photon per spectral bin[58]. The power-coupling coefficient $\theta$ was swept to produce the map in Fig. 4a. In addition, to account for the detuning, $\delta_0$, between the pump laser wavelength and the resonance center, a corresponding phase shift was applied to the current intracavity field. The propagation simulation was performed using a split-step nonlinear Schrödinger equation solver, where the linear step (with fixed power loss parameter $\alpha = 0.1$ dB/cm, corresponding to an intrinsic quality factor of 1.8 million, and swept group velocity dispersion coefficient, $\beta_2$) and the nonlinear step (with fixed nonlinear parameter $\gamma = 2$ W/m) were performed iteratively in 16 steps along the ring.

To ensure that the comb state converged to a dark-pulse comb, the intracavity field was initialized with a square wave whose upper and lower power values correspond to the continuous-wave bistability solutions[33]. Once the field inside the microresonator had converged to a steady state, the result was analyzed. For each combination of $\beta_2$ and $\theta$, the detuning parameter and the initial square pulse width giving the best comb state were chosen. Although there are several metrics by which combs can be evaluated, in this work, we optimized the parameters for maximum power in the weakest line within the C band[31].

**Data availability.** The code (including the DSP scripts) and raw data necessary to reproduce the plots in this work can be accessed at https://doi.org/10.5281/zenodo.1206122.

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

## Acknowledgements

The work in this publication was supported by the Swedish Research Council (VR); the National Science Foundation (NSF) (1509578-ECCS); DARPA (W31P4Q-13-1-0018); and AFOSR (FA9550-15-1-0211).

## Author contributions

A.F., M.M., and A.L.-R. carried out the transmission experiments. M.M. and A.L.-R. designed the transceiver subsystems with back-to-back characterization. A.F. processed the data, characterized the dark-pulse comb, and ran dispersion simulations. Ó.B.H. performed the comb bandwidth simulations. Y.X. and M.Q. designed and manufactured the ring resonators. P.-H.W. and D.E.L. performed initial device verification. All authors discussed the results. A.F. wrote the first draft of the paper. P.A.A., A.M.W., and V.T.-C. supervised the work.
