## [Peer Review File · Nature Communications]

Reviewers' comments:

Reviewer #1 (Remarks to the Author):

1. Major claims of the paper

In this publication by A. Fülöp et al. the authors demonstrate for the first time data transmission with a dark-pulse Kerr comb generated in a microresonator. In this work, they claim the following:

a) First transmission experiment using the Kerr frequency comb of a dark pulse state, generated in a SiN microresonator, as optical source.

b) Data transmission at 4.4 Tbit/s using 20 carriers of the frequency comb separated by 230 GHz. Highest order modulation format, 64QAM, used so far with any integrated comb source.

c) Conversion efficiency, pump to comb, above 20%.

d) Optimized microresonator's parameters leading to maximum OSNR, ideal for applications in optical communication.

2. Novelty and broad impact:

This is the first time that a dark pulse, only recently observed in microresonators, is used for data communications. The authors exploit the capability of frequency comb generation in normal dispersion regime to relax the fabrication requirements needed for the generation of bright combs, and providing, in addition, higher conversion efficiency.

This, not only approves dark comb states as potential candidates for optical sources in fully integrated transmitters, but also represents a step forward on the understanding of Kerr comb states, over-all, in microresonators. This has become, in very short time, a subject of remarkable intense study by groups of very diverse communities and source of many well renowned publications. Thus, I believe this publication has the potential to impact both scientifically and technologically.

3. Validation of the claims listed in section 1:

a) The authors depict the intracavity pulse measured following two different techniques but giving similar results. The techniques are well described and referenced in the methods, and the pulse shape fits with that of a dark pulse. Thus, proving their claim.

b) The transmission experiment is well detailed, with both text and figures. The transmission and reception system for proof of claim uses the standard odd/even separation of the carriers for WDM emulation, split-and-delay technique for emulation of polarization multiplexing and individual channel reception. Thus, I do not detect any flaws in their description and results.

A close observation of Fig. 2c shows what seems to be plateaus of low optical power for one set of the channels. At first glance, this could be related either to residual AWGN which is filtered by the WSS, or to Nyquist-like pulse shaping together with a strong not modulated carrier. Please, specify the reason for the diverse channel spectra.

Please specify, if any, the pulse shaping.

c) I cannot see flaws in their conversion efficiency claim. First, the authors provide all necessary power levels. Second, the claimed conversion efficiency is consistent with those of ref. 35 and 38.

d) Optimizing the power of the weakest line of the comb will directly influence the minimum OSNR. The authors swept the coupling value and dispersion parameter. The authors should comment on the technical viability of the coupling and dispersion values, which maximizes the power of the weakest line.

Please, indicate the power conversion efficiency and the flattened net conversion efficiency of the simulated comb for optimized parameters.

4. Reproducibility

Given the high level of detail provided by the authors, both in the manuscript and in the raw data online, I believe the experiments are reproducible in other laboratories, provided similar hardware is used.

5. Clarity of text, methods and images:

I approve the authors' distribution of information between the main text and the Methods. This leads to a clean and easy to follow manuscript, well supported by the more technical information described in the Methods. Images are neat and relevant.

To slightly improve the technical accuracy, I would suggest re-phrasing Line 71: "... 20 channels modulated using..." . Channels are, strictly speaking, not modulated.

6. References:

All related papers have been correctly recognized and references are well cited.

If necessary, additional references may be added following the points in section 7.

7. Additional queries/suggestions:

The authors' comb source features a drop port, which is solely used for intracavity pulse characterization and verification of circulating dark pulse.

a) Please, comment how removing the drop port would alter relevant parameters such as power conversion efficiency and spectral envelope of the comb

b) Have the authors investigated further microresonators? Please, discuss how reproducible are the claimed power conversion efficiency and flattened net conversion efficiency for other equivalent microresonators. What is the percentage of microresonators leading to a dark pulse state compatible for data transmission applications?

The authors show in Fig 4c the theoretical OSNR required for different modulation formats and symbol rates. Symbol rates are 28 GBd and 56 GBd. For an efficient use of the spectrum, the symbol rate is typically quite close to the FSR of the comb. However, the authors simulated an optimized comb with an FSR of 100 GHz. Furthermore, the authors cite a recent work, Ref. 51, proving the viability of 50 GHz space comb from a dark pulse.

c) Please, provide the reason for the selected symbol rates and FSR.

d) The bandwidth of conventional transmitters and receivers would strongly limit the OSNR of channels with high symbol rates, leading to low spectral efficiency when using a 100 GHz FSR comb. Thus, the 50 GHz FSR comb, ref. 51, would potentially lead to higher spectral efficiency. Please, discuss how a larger resonator, providing 50 GHz FSR, would change the claims retrieved from the simulation, e.g., would PM-64QAM still be possible?

The authors use for their experiments and off-chip pump power of 25.6 dBm. This would lead to ~ 100 mW coupled to the microresonator (5 dB per facet). Later on, they consider a 100 mW pump power for their simulations, which, as specified in the Methods, is coupled to the microresonator.

e) Please, discuss the OSNR change when decreasing the input power to the microresonator.

Authors could provide the minimum pump power, which would still allow 64QAM transmission of C band channels.

f) Please, discuss if higher modulation formats, e.g. 256QAM, or higher symbol rates, e.g. 56 GBd, would still be compatible with an integrated laser providing 100 mW pump power.

Finally, the authors envision a "fully integrated transmitter system featuring an optimized dark-pulse covering the whole C band":

g) Please, comment on the possibility to obtain a frequency comb from a dark pulse covering the L-band and if shifting to larger wavelength imposes limitations to dark pulse generation.

h) The authors should provide the laboratory conditions in which the microresonators were operated, e.g., temperature and humidity, and how these conditions compare to those where such transmitters would be utilized.

8. Recommendation

I believe the novelty and strength of the claims together with the broad impact qualifies the paper for the wide audience standards of Nature Communications. I can recommend its publication once the points listed in the previous sections are well addressed.

Reviewer #2 (Remarks to the Author):

In this paper, the authors demonstrate coherent communications using high power dark pulse Kerr combs. Compared to previous work, e.g., ref. 31 where 16QAM modulation was used, they achieved 64 QAM modulation. This improvement is enabled by the high conversion of the dark pulse Kerr combs. Although the achieved aggregate bit rate is lower than ref. 31, the demonstration of high order modulation format is useful for further boosting the data transmission rate in the future. The paper is clearly written and the results are very solid and interesting. It has the potential to be accepted to Nature Communications, however, there are several technical issues that need to be addressed:

1) The authors mentioned that third order dispersion is responsible for the generation of the asymmetric comb in Fig. 1b (line104). However, the bandwidth is not so broad, which makes the third order dispersion may not be so important. Other factors causing this asymmetry may include Raman, mode-interaction. So this argument is not well justified.

2) When flattening the comb spectrum, the useful efficiency drops a lot from 20% to 1.5% (line122). This makes the high efficiency comb much less attracting. Can we mitigate this efficiency drop?

3) Please specify the data rate (GBd) used when testing the BER in the caption of Fig. 3. Also, I suggest to provide the achieved aggregate data rate in the Introduction section, when summarizing the content.

4) When using high order modulation, the high power per line is important; the stability performance of the comb lines is also important. More discussion on this performance should be covered.

5) For the simulations in Fig. 4, please give the detuning used in the Method section. More importantly, the dark pulses are found to have multi-stability in simulations e.g., supplementary information of ref. 33 and Parra-Rivas, P., Knobloch, E., Gomila, D., & Gelens, L. (2016). Dark solitons in the Lugiato-Lefever equation with normal dispersion. *Physical Review A*, 93(6), 063839. This means for a same pump power and detuning and cavity parameters (such as coupling rate and dispersion), dark pulses with different waveform shape can be stable simultaneously. Then, it means that the output comb shape can be quite different even for a same pump condition and a same cavity parameter. The authors should confirm that the output is always the same even using different initial conditions, otherwise the results in Fig. 4 are not so useful.

Reviewer #3 (Remarks to the Author):

The main claim of this paper is the modulation of a 200GHz spaced micro-resonator comb with 64QAM formatted data at a symbol rate of 20 Gbaud. Emphasis is placed specifically on an integrated comb operating at 64QAM. There is sufficient detail for the experiment to be reproduced given access to suitable components. The experiment has been carried out to a high standard, and

the results are convincing, enabling the efficacy of the approach to be assessed.

In my opinion the novelty of this paper is significantly below what one expects for this journal, the performance of similar comb lasers has been previously reported [33-35], including reports of this laser [38-39]. Modulation of comb sources at low spectral efficiency (as reported here) is also well known (an incomplete list is provided [1-7, 23-26]). A comb laser is useful if it provides enhanced performance compared to an array of lasers, for example enhanced frequency stability. The tests performed here provide hardly any verification of important comb parameters such as the accuracy and stability of the comb line spacing. For example operation with Nyquist spaced channels would reveal penalties from any unstable line spacing [2]. A result such as that presented here is of course of some interest to the specialist community interested in using combs for communications, but due to the poor spectral efficiency communicating to this audience is more suited to a conventional engineering journal. For the wider community, there is nothing reported here which has not been previously apparent.

A statistical analysis is of relevance to the results in figure 3, and is conspicuous by its absence. A reader is left wondering for themselves if the variations are caused directly by variations in the comb parameters (a parametric analysis by the authors would confirm this) or simple statistical variations owing to the small sample sizes available to researchers performing off-line processing (a simple statistical analysis by the authors would equally confirm this conclusion).

The conclusion clearly states "We have shown the highest-order modulation format demonstrated using

180 any integrated comb source". The authors have certainly demonstrated 64QAM, but not for the first time [2]. They have demonstrated a partially integrated solution (partially because an EDFA, filter and separate tunable laser with feedback are required in the transmitter). However, partially integrated solutions have been reported before [DOI: OE.25.017847, DOI: 10.1109/CLEOE-EQEC.2017.8086954, <http://ieeexplore.ieee.org/document/7937050/citations>]. The paper makes no demonstration of the advantage of the particular of frequency comb used with respect to these, and other, alternatives. The paper omits transmission over any appreciable distance which also detracts from its value. The paper simply combines two specific previously known technologies in a way in which alternative technologies meeting the same objectives have been combined previously. Therefore this paper does not represent any increase in understanding likely to influence thinking in the field, and is much more suited to a specialist journal.

Response to review comments for Nature Communications submission “High-order coherent communications using mode-locked dark-pulse Kerr combs from microresonators”

We thank all the reviewers for their time and for their exhaustive and critical analysis of our work. Please let us detail our response to the concerns in their report. The changes in the manuscript are highlighted in **yellow**. Please note that we now include new Supplementary material.

Reviewer 1

Comment: 3 b). The transmission experiment is well detailed, with both text and figures. The transmission and reception system for proof of claim uses the standard odd/even separation of the carriers for WDM emulation, split-and-delay technique for emulation of polarization multiplexing and individual channel reception. Thus, I do not detect any flaws in their description and results. A close observation of Fig. 2c shows what seems to be plateaus of low optical power for one set of the channels. At first glance, this could be related either to residual AWGN which is filtered by the WSS, or to Nyquist-like pulse shaping together with a strong not modulated carrier. Please, specify the reason for the diverse channel spectra. Please specify, if any, the pulse shaping.

Response: The reviewer is correct in that the residual AWGN filtered by the WSS is the cause for the noise plateaus. The initial comb flattening was done individually for the even and the odd arms (at the two outputs of the first WSS). While the noise level entering the initial WSS was flat (Fig. 2b), the separate flattening translated to somewhat different power levels in the two arms. Combined with the gain tilt of the EDFAs inside the modulator section, these effects mandate a second flattening procedure in the second WSS. This ensures that the power level of all channels remains equal when entering the fiber span. The WSS was selectively attenuating each channel (and the region around it) as much as was needed, resulting in the varying noise floor in Fig. 2c. No pulse shaping was applied in the AWG.

Action taken: The sentences describing the transmitter in the subsection “Optical data modulation” have been adjusted slightly with the following added information: “The WSS is also used to equalize the powers among the comb lines in each arm separately.” The main text now includes the following sentence: “The second flattening will translate the power differences in the two arms (caused by the comb envelope and initial flattening step) into the slightly varying noise floor seen in Fig. 2c”. That pulse shaping is not used is now explicitly mentioned in the main text: “The AWG is programmed to generate two independent random 64QAM signals using square pulses ...”.

Comment: 3 d). Optimizing the power of the weakest line of the comb will directly influence the minimum OSNR. The authors swept the coupling value and dispersion parameter. The authors should comment on the technical viability of the coupling and dispersion values, which maximizes the power of the weakest line.

Please, indicate the power conversion efficiency and the flattened net conversion efficiency of the simulated comb for optimized parameters.

Response: The swept regions of dispersion are reachable with rectangular-shaped waveguides, this is now described in more detail in the new Supplementary note 1. The coupling constant can in principle be adjusted by varying the distance between the bus waveguide and the ring. Should the fabrication process not allow for very narrow gaps, a pulley-coupling scheme can be applied instead. The pulley-coupling scenario is also analyzed in detail and verified in Supplementary note 1.

Actions taken: The following has been added to the discussion: “(the fabrication feasibility of these parameters is discussed and verified in Supplementary note 1)”. The figure text for Fig. 4 has been updated to mention the power conversion efficiency as well as the flattened net conversion efficiency: “The comb spacing is fixed at 100 GHz. The peak value marked with X at -10 dBm is achieved for $\beta_2=350$ ps²/km, $\theta=0.011$ and a detuning of $\delta_0=0.0445$ yielding a comb with above 50% power conversion efficiency and a flattened net conversion efficiency of 4.4%”. Supplementary note 1 has been added and it discusses in detail the feasibility of optimized dark pulse combs.

Comment: 5. To slightly improve the technical accuracy, I would suggest re-phrasing Line 71: “ ... 20 channels modulated using...” . Channels are, strictly speaking, not modulated.

Action taken: The sentence has been adjusted: “Using off-chip pump powers below 400 mW, we demonstrate 80 km data transmission with 20 channels. Each channel contains data modulated using 20 GBd 64-quadrature amplitude modulation (QAM) resulting in an aggregate data rate of 4.4 Tb/s.”

Comment: 7. The authors’ comb source features a drop port, which is solely used for intracavity pulse characterization and verification of circulating dark pulse.

a) Please, comment how removing the drop port would alter relevant parameters such as power conversion efficiency and spectral envelope of the comb.

Response: As the drop port is only weakly coupled, its presence only has minor impact on the intracavity field. The gap between the ring and the drop waveguide is significantly larger (1000 nm for the drop port vs. 300 nm for the through port). According to the coupling simulations included in Supplementary note 1, the coupling strength is therefore expected to be more than 10 times weaker. This means that power losses owing to light getting coupled out of the resonator are dominated by the through port.

Action taken: The new Methods subsection “Microresonator operation details” now contains information on the drop port parameters: “On the chip, the coupling between the bus waveguide and the ring was set by their 300 nm wide gap. As the drop port gap (at 1000 nm) was much larger, the power coupling to that port is estimated to be more than 10 times weaker (see Supplementary note 1). The roundtrip losses owing to light getting coupled out of the resonator are therefore expected to be dominated by the through port.”

Comment: b) Have the authors investigated further microresonators? Please, discuss how reproducible are the claimed power conversion efficiency and flattened net conversion efficiency for other equivalent

microresonators. What is the percentage of microresonators leading to a dark pulse state compatible for data transmission applications?

Response: The comb shape and initialization strongly depend on the presence of modal interactions that are at present difficult to precisely control (see ref. 52 in the revised manuscript), therefore the exact spectral envelope is difficult to reproduce. Out of three available devices with similar geometries, two produce dark pulse combs when pumped at the 1540 nm resonance with different resulting spectral envelopes. Several devices with comparable conversion efficiencies and qualitatively similar spectra have however been produced and reported (as in refs. 33 and 35). An alternative to produce microresonator combs with controllable mode interaction and line spacing is presented in ref. 53. This technology may prove useful to generate dark pulse combs but this prospect requires further investigation.

Action taken: The following sentences have been added to the new “Microresonator operation details” Methods subsection: “As the comb generation process depends strongly on the presence of modal coupling⁵², the spectral envelope is sensitive to slight fabrication variations. Out of three similar devices, two produce dark pulse combs using a 1540 nm pump. The tolerances (and thereby the device yield) is however expected to be improved using techniques enabling post-fabrication tuning, for example using controllable mode interactions as described in ref. 53.”

Comment: The authors show in Fig 4c the theoretical OSNR required for different modulation formats and symbol rates. Symbol rates are 28 GBd and 56 GBd. For an efficient use of the spectrum, the symbol rate is typically quite close to the FSR of the comb. However, the authors simulated an optimized comb with an FSR of 100 GHz. Furthermore, the authors cite a recent work, Ref. 51, proving the viability of 50 GHz space comb from a dark pulse.

c) Please, provide the reason for the selected symbol rates and FSR.

Response: The reviewer is right that the symbol rates should more closely match the FSR when taking the spectral efficiency metric into account. The plots have now been replaced with 50 GBd and 100 GBd figures to give a better upper benchmark matching the ring line spacing in the simulations.

Action taken: The figure has been changed to show the requirements for 50 GBd and 100 GBd symbol rates instead. A new supplementary note was added containing simulations of 50 GHz and 100 GHz spaced dark pulse combs spanning C and C+L-bands.

Comment: d) The bandwidth of conventional transmitters and receivers would strongly limit the OSNR of channels with high symbol rates, leading to low spectral efficiency when using a 100 GHz FSR comb. Thus, the 50 GHz FSR comb, ref. 51, would potentially lead to higher spectral efficiency. Please, discuss how a larger resonator, providing 50 GHz FSR, would change the claims retrieved from the simulation, e.g., would PM-64QAM still be possible?

Response: Dark pulse combs can in principle be designed to fit many different scenarios in terms of number of lines and line spacing. As we can now see in the new supplementary note 2, increasing the number of generated lines will however come at the expense of lowering the minimum line powers. The simulation results show that a 50 GHz C-band spanning comb is similar to a 100 GHz C+L band spanning

one, indicating that the conversion efficiency and lowest line power is mainly determined by the number of generated lines. Going from a 100 GHz C-band comb to a 50 GHz one yields an 8 dB lower minimum line power. While this will decrease the available OSNR margin, data modulation using PM-64QAM should remain possible.

Action taken: In Supplementary note 2 we now describe the C-band and C+L-band situations for 50 GHz and 100 GHz combs.

Comment: The authors use for their experiments and off-chip pump power of 25.6 dBm. This would lead to ~ 100 mW coupled to the microresonator (5 dB per facet). Later on, they consider a 100 mW pump power for their simulations, which, as specified in the Methods, is coupled to the microresonator.

e) Please, discuss the OSNR change when decreasing the input power to the microresonator. Authors could provide the minimum pump power, which would still allow 64QAM transmission of C band channels.

Response: Simulations included in Supplementary note 2 (particularly Fig. S3b) display the relationship between the achievable weakest line power and the pump power. Within the evaluated region (pump powers between 25 mW and 200 mW) decreasing the pump power by 3 dB lowers the achievable weakest line power by 4-6 dB. Even for a 100 GHz C-band comb pumped with 25 mW, the weakest line remains at around -20 dBm (corresponding to 38 dB above the quantum noise floor). The available OSNR would in principle allow this comb to carry 64QAM channels. It is worth noting however that the limit of when high order transmission becomes possible will depend significantly on the placement and penalties of the surrounding components, choice of FEC limit, and the transmission distance.

Action taken: New simulations have been added to Supplementary note 2. They describe comb variations for 50 GHz and 100 GHz combs with pump powers between 25 mW and 200 mW.

Comment: f) Please, discuss if higher modulation formats, e.g. 256QAM, or higher symbol rates, e.g. 56 GBd, would still be compatible with an integrated laser providing 100 mW pump power.

Response: A system permitting 256QAM on all comb lines is challenging with the chosen FEC threshold and 100 mW pump power. However, as discussed now in the Supplementary note 3, it is possible to design a transmitter architecture with unequal OSNR (but equalized power) between channels. The central lines display high OSNR and could allow for 256QAM assuming that the modulators and data sources for each channel can be configured individually.

Action taken: The following sentence has been added to the discussion section: "Given the high available power in the central lines, these can in principle carry data using even higher order modulation formats, potentially targeting PM-256QAM. A transmitter scheme where the power in the stronger lines can be exploited in this manner is discussed in Supplementary note 3".

Comment: Finally, the authors envision a "fully integrated transmitter system featuring an optimized dark-pulse covering the whole C band":

g) Please, comment on the possibility to obtain a frequency comb from a dark pulse covering the L-band and if shifting to larger wavelength imposes limitations to dark pulse generation.

Response: Owing to silicon nitride's large transparency window, going to slightly longer wavelengths, such as the L-band is definitely possible. Combs covering both C and L-bands will require more lines and will therefore come at a cost with respect to their minimum powers and the conversion efficiency (similarly to what is stated in ref. 36).

Action taken: Supplementary note 2 now includes dark pulse combs covering the C+L bands.

Comment: h) The authors should provide the laboratory conditions in which the microresonators were operated, e.g., temperature and humidity, and how these conditions compare to those where such transmitters would be utilized.

Response: The temperature of the positioning stage was kept stable at 18°C (less than 0.01°C variation) using a standard laser temperature controller. Alternatively, the temperature control could be done with an integrated thermo-optic heater. There were no special efforts made to keep the laboratory humidity stable beyond standard air conditioning, so this parameter was therefore not measured.

Action taken: The new "Microresonator operation" Methods subsection contains this information: "The chip containing the microresonator was kept on a piezo-controlled positioning stage stabilized using a standard laser temperature controller at 18°C with less than 0.01°C variation."

Reviewer 2

Comment: 1) The authors mentioned that third order dispersion is responsible for the generation of the asymmetric comb in Fig. 1b (line104). However, the bandwidth is not so broad, which makes the third order dispersion may not be so important. Other factors causing this asymmetry may include Raman, mode-interaction. So this argument is not well justified.

Response: We have calculated the propagation constant for our designed geometry using a mode solver. Using this data, we can simulate the comb generation process as described in the Methods section. The numerical results (see parameters in the inset) are compared to the experimentally measured comb (Fig. 1b in main text):

These results indicate that our simulations correctly predict most of the salient features of the comb in the spectral domain. However, it is true that even when including the full dispersion (i.e. third and higher order terms) of the waveguide mode, the expected comb envelope is slightly more symmetric than the measured one. Possible reasons for this slight discrepancy might be Raman effect or a slight distortion in dispersion due to linear coupling between modes. We agree with the reviewer that the claim is therefore not duly justified.

Actions taken: The sentence mentioning that third order dispersion is responsible for the asymmetry of the comb has been removed.

Comment: 2) When flattening the comb spectrum, the useful efficiency drops a lot from 20% to 1.5% (line122). This makes the high efficiency comb much less attracting. Can we mitigate this efficiency drop?

Response: The practical efficiency drop can in principle be counteracted by making use of the increased powers (and the resulting higher OSNRs) in the central lines by for example modulating data with higher-

order modulation formats on them or using different coding rates. In a laboratory scenario where we are limited in the number of modulators and signal generators this cannot easily be achieved. The supplementary section now contains an example sketch suggestion for such a setup. This transmitter architecture makes use of the conversion efficiency from dark pulse combs.

Actions taken: Supplementary note 3 has been added and referenced in the discussion section.

Comment: 3) Please specify the data rate (GBd) used when testing the BER in the caption of Fig. 3. Also, I suggest to provide the achieved aggregate data rate in the Introduction section, when summarizing the content.

Actions taken: The caption to Fig. 3 now includes “All measurements were performed with random data modulated using 20 GBd PM-64QAM” while the introduction has been adjusted slightly: “Each channel contains data modulated using 20 GBd 64-quadrature amplitude modulation (QAM) resulting in an aggregate data rate of 4.4 Tb/s”.

Comment: 4) When using high order modulation, the high power per line is important; the stability performance of the comb lines is also important. More discussion on this performance should be covered.

Response: We have previously verified that the feedback loop ensures a stable average power with variation <0.1 dB over several hours (see ref. 38). We now present here an analysis of the line spacing stability of the comb using electro-optic downconversion. The results clearly indicate stable mode-locking. This level of line spacing stability is not necessary for the measurements presented in this work because the WDM channels are largely spaced. However, these new results indicate that the comb has sufficient stability to allow for densely packed WDM, assuming the challenges associated when moving towards narrower FSRs are overcome.

Action taken: New measurements characterizing the line spacing stability have been performed. The results are now included in Fig. 1e while the measurement is detailed in a new section in the Methods.

Comment: 5) For the simulations in Fig. 4, please give the detuning used in the Method section. More importantly, the dark pulses are found to have multi-stability in simulations e.g., supplementary information of ref. 33 and Parra-Rivas, P., Knobloch, E., Gomila, D., & Gelens, L. (2016). Dark solitons in the Lugiato-Lefever equation with normal dispersion. *Physical Review A*, 93(6), 063839. This means for a same pump power and detuning and cavity parameters (such as coupling rate and dispersion), dark pulses with different waveform shape can be stable simultaneously. Then, it means that the output comb shape can be quite different even for a same pump condition and a same cavity parameter. The authors should confirm that the output is always the same even using different initial conditions, otherwise the results in Fig. 4 are not so useful.

Response: Against all tested initial conditions, our algorithm finds the steady-state solution that maximizes the power of the weakest comb line. Varying the noise seed does not impact the steady state that the solution converges to. While the initializing square pulse width does influence the final state

(similar to ref. 33), the system map in Fig. 4a describes the best stable state given the system parameters.

Action taken: The detuning is now mentioned in the caption of Fig. 4 as well as in the new Supplementary note 2. The methods section describing the simulations now mentions the optimization of the initial square wave pulse: “For each combination of β_2 and θ , the detuning parameter and initial square pulse width giving the best comb state was chosen”.

Reviewer 3

Comment 1: In my opinion the novelty of this paper is significantly below what one expects for this journal, the performance of similar comb lasers has been previously reported [33-35], including reports of this laser [38-39]. Modulation of comb sources at low spectral efficiency (as reported here) is also well known (an incomplete list is provided [1-7, 23-26]).

Response: The existence of dark-pulse comb states has been reported previously (refs. 33-35), but this is however the first time they have been used in an optical communication context (the conference proceeding in ref. 38 only deals with the stabilization vaguely hinting at the application while ref. 39 is a different device for which dark pulse states have not been realized, focusing on the fabrication details and the observation of unusually high quality factors). Our work represents a milestone in the field because we show that the net conversion efficiency of the comb source allows for modern coherent optical communications while keeping pump power level compatible with integrated silicon sources. In addition, we now report (see replies to reviewers #1 & #2) that level of line spacing stability of this type of comb is significantly better than what can be obtained with independent tunable laser sources. Our manuscript also investigates the fundamental scaling of the conversion efficiency of dark pulse combs with the bandwidth, and discusses further manufacturability challenges. These aspects are not detailed in previous publications.

We also acknowledge that there is more research to be done, especially with regards to maximizing device yields and optimizing the initialization of dark-pulse comb sources. We would nevertheless like to argue that the work here presented constitutes a significant step towards an envisioned fully integrated multi-channel comb-based transmitter.

Comment 2: A comb laser is useful if it provides enhanced performance compared to an array of lasers, for example enhanced frequency stability. The tests performed here provide hardly any verification of important comb parameters such as the accuracy and stability of the comb line spacing. For example operation with Nyquist spaced channels would reveal penalties from any unstable line spacing [2]. A result such as that presented here is of course of some interest to the specialist community interested in using combs for communications, but due to the poor spectral efficiency communicating to this audience is more suited to a conventional engineering journal. For the wider community, there is nothing reported here which has not been previously apparent.

Response: We have now performed characterization of the line spacing stability of our device using electro-optic downconversion. The stability exceeds what is easily achievable using standard external cavity lasers. This level of stability was not necessary for our experiment owing to the large line spacing however. In a narrower spaced comb situation (such as with the 50 GHz case in ref. 51) where spectral efficiency maximization becomes a possible target, this level of stability could allow for densely packed channels as well as advanced pre-compensation algorithms (such as in ref. 8).

Action taken: The frequency stability is now mentioned in the comb description with the associated part of the methods section describing the measurement.

Comment 3: A statistical analysis is of relevance to the results in figure 3, and is conspicuous by its absence. A reader is left wondering for themselves if the variations are caused directly by variations in the comb parameters (a parametric analysis by the authors would confirm this) or simple statistical variations owing to the small sample sizes available to researchers performing off-line processing (a simple statistical analysis by the authors would equally confirm this conclusion).

Response: Variations in bit error ratio on the order of factors of 2-3 are expected for high order modulation format demonstrations and are in line with similar experiments carried out in our lab with a different, more standard comb source (see e.g. Mazur et al. European Conference on Optical Communication 2017, p. M1F5 for similar variations using an electro-optic comb source). Both transmitter and receiver components are expected to have some small wavelength-dependent behavior that becomes very visible in the BER metric. To verify that the statistics within each wavelength recording is good, we have now updated Fig. 3c. It now includes error bars showing BERs for the best and the worst batch (out of a total of 5 for each wavelength). Each batch includes more than 9 million transmitted bits (as is specified in the DSP part of the Methods section).

Action taken: Figure 3c has been updated to include error bars. The figure text has also been expanded: “The error bars show the BER values for the best and the worst batch for each wavelength. The wavelength-dependent BER variations are within expected levels considering the transceiver components”.

Comment 4: The conclusion clearly states “We have shown the highest-order modulation format demonstrated using 180 any integrated comb source”. The authors have certainly demonstrated 64QAM, but not for the first time [2]. They have demonstrated a partially integrated solution (partially because an EDFA, filter and separate tunable laser with feedback are required in the transmitter). However, partially integrated solutions have been reported before [DOI: OE.25.017847, DOI: 10.1109/CLEOE-EQEC.2017.8086954 ,<http://ieeexplore.ieee.org/document/7937050/citations>]. The paper makes no demonstration of the advantage of the particular of frequency comb used with respect to these, and other, alternatives.

Response: While the solution is indeed only integrated partially, the demonstrated comb source provides a clear pathway towards further integration. The amplified laser in this demonstration results in an on-chip pump power compatible with state-of-the-art chip-scale lasers, making both the EDFA and filter only necessary in the present lab environment. At the same time, the comb emits high-enough powers in 20 lines to make them compatible with high-order modulation formats. Supplementary note 2 now includes an analysis of an optimized comb featuring high OSNR per line while keeping the pump powers < 100 mW. The solutions described in the reviewer comment all contain components that, in our view, make them less favorable for near-future integration (1st and 2nd: mode-locked laser with an external highly nonlinear fiber-based broadening which requires amplifiers and filters. The filters based on Brillouin amplification further require both electro-optic modulation and more amplification, 3rd: electro-optic modulation which requires high-power high-frequency microwave components). These solutions are of course worth exploring, and we have acknowledged on-going efforts in this direction in the introduction (see refs. 13-16) that represent state of the art partially integrated comb sources based

on mode-locked lasers and electro-optic frequency combs. Based on the results presented in this manuscript, we argue that dark-pulse combs can enable multi-channel optical transmitters with significantly fewer roadblocks remaining.

Comment 5: The paper omits transmission over any appreciable distance which also detracts from its value. The paper simply combines two specific previously known technologies in a way in which alternative technologies meeting the same objectives have been combined previously. Therefore this paper does not represent any increase understanding likely to influence thinking in the field, and is much more suited to a specialist journal.

Response: The 80 km single-span demonstrated distance proves that the system allows more than just back-to-back performance. The distance is similar to other comb-based demonstrations (eg. 16QAM in ref. 31, 32QAM in 10.1364/OFC.2017.Th5C.3, 64QAM in 10.1364/OFC.2017.Th3F.4, and 128QAM in 10.1109/JLT.2017.2786750), as performance implications beyond the first span vary in a well-studied manner and depend less on the initial light source and more on the transmission link implementation. By selecting a BER limit compatible with hard-decision decoding, we have left ample margin for extending the reach using for example soft-decision codes. We argue that the main feature in this paper is the possibility to attain high-OSNRs and stable line spacing while keeping a pump power level compatible with state of the art hybrid silicon lasers. This has allowed us to reach very high modulation formats. The manuscript now provides in addition a careful look into manufacturability and the fundamental scaling of the power conversion efficiency of dark pulse combs. We therefore think that the content presented in this paper has the potential to reach and influence the field beyond what can be reached in a specialist journal.

Reviewers' comments:

Reviewer #1 (Remarks to the Author):

The authors have properly addressed my requests by respectively modifying/adding new information into the manuscript.

The authors have added new Method sections, namely "Line spacing stability measurements" and "Microresonator operation". The later makes the manuscript more transparent, allowing the reader to understand the details behind the optical source and its operation. I believe these two new sections consolidate the content of the manuscript. In addition, the authors include Supplementary material.

Regarding the modifications, I have just two remarks:

- Line 73: the authors mention that the "Aggregate data rate" is 4.4 Tbit/s. In fact, that is the "Aggregate net data rate", or simply "net data rate", which considers the overhead. Indeed, I would suggest the following: Provide in line 73 the aggregate data rate, or line rate, which is 4.8 Tbit/s, rather than the net data rate, due to the fact that no overhead is yet declared. Later on, when the overhead is specified, provide the aggregate net data rate, or simply net data rate, possibly together with the aggregate data rate. This should avoid possible confusions.
- Line 72: I would suggest to replace "data modulated" by "data encoded"

I cannot see any flaws in their simulations and explanations. Thus, I recommend publication of the manuscript.

Reviewer #2 (Remarks to the Author):

The authors have addressed the points in my review in their responses and with revisions to the manuscript. I recommend publication of the revised manuscript.

Reviewer #3 (Remarks to the Author):

1. Reviewer 3's opinion of response to reviewer 3.

Comment 1; The authors agree with the general sentiment that the work reported here should be judged primarily on the system results: "Our work represents a milestone in the field because we show that the net conversion efficiency of the comb source allows for modern coherent optical communications while keeping pump power level compatible with integrated silicon sources". I am not qualified to judge the second claim, but the other two reviewers appear to be satisfied with this claim. The claim for modern coherent optical communications remains incorrect.

- Modern communications which would benefit from a comb source operate with higher spectral efficiencies, almost to the point where slightly exceeding the frequency stability of modern ITLAs would give a problem (a few GHz guard band). A guard band of 80 GHz is simply nearly 20 years out of date.

- Modern commercially available systems already operate at 600 Gbit/s per channel [<https://doi.org/10.1364/JOCN.9.000C12>] using 12 coded bits per symbol (PM-64QAM) and a symbol rate of 56 Gbaud. The net information spectral density of 600 Gbit/s of commercial systems is 10 bit/s/Hz (dual polarisation). This paper only presents results at 3.6 bit/s/Hz, approximately 1/3rd of commercially available systems.

For a specialist journal where one may focus on optimisation of certain parameters, the results remain sufficient for publication. However, in terms of generating a "milestone in the field" I am afraid I disagree with this authors. In my opinion this paper does not represent a milestone in

modern coherent optical communications. The device level measurement of beat note linewidth does not add or detract from the milestone nature of the systems result.

Comment 2; The authors have now added an appropriate measurement to address this point. The beat note bandwidth of 30kHz is indeed lower than that which would be observed from some free running lasers, but is significantly higher than that which is observed on other comb sources. I disagree with the author's conclusions from this data, but sufficient data has been presented for me to come to my own opinion so this is OK. However the true test of the suitability of the laser to the proposed application is a test of the system performance at a high spectral density, either directly or by emulation. Therefore I do not feel that the authors have done enough to address my concern to merit publication in this journal, although the new results make the manuscript even more suited to a specialist journal.

Comment 3; This has been adequately addressed.

Comment 4; I welcome the authors' acknowledgment of other works in the introduction, this is helpful to the reader. However, a source is either fully integrated and so only requires one fibre pigtail and fits inside a semiconductor laser package, or it is not fully integrated and requires multiple pigtails (for the device and for other components) and the eventual package would more resemble that of an EDFA. The author restating their claims that this has a few attractive characteristics does little to convince me that a fully integrated transmitter is enabled by this technology.

Comment 5; I would agree with the authors that "the main feature in this paper is the possibility to attain high-OSNRs and stable line spacing while keeping a pump power level compatible with state of the art hybrid silicon lasers". However, this type of excellent engineering development is more suited to a specialist journal. The authors acknowledge that they have made no attempt beyond this to influence thinking in terms of overall system design.

2. Reviewer 3 opinion on response to reviewer 1.

The reviewer's comments seem to have been addressed adequately to me.

It is interesting to note that the authors have acknowledged that a 100 GHz comb would find its best application in a 100 Gbaud (or so) system in their response "The reviewer is right that the symbol rates should more closely match the FSR". I am afraid the fact that the experiment is a long way away from this ideal is one of the reasons why I cannot regard this paper as a milestone.

3. Reviewer 3 opinion on response to reviewer 2.

Most of the reviewer's comments seem to have been addressed adequately to me, except;

Comment 4: As the reviewer states "When using high order modulation...the stability performance of the comb lines is also important. More discussion on this performance should be covered". The authors have provided a measurement but do not discuss or experimentally demonstrate what levels of frequency jitter and what types of jitter spectrum are required (their beat note linewidths do not look Lorentzian).

Overall, the device work here is of very high quality and merits publication in a specialist journal as it represents an enhancement of the performance through careful engineering of the design and excellent fabrication. The "milestone nature" and wider appeal rest on the system results, and whilst these are performed well, in this reviewer's opinion, they do not represent a significant milestone due to the very large guard bands employed.

Response to review comments for Nature Communications submission “High-order coherent communications using mode-locked dark-pulse Kerr combs from microresonators”

We thank the reviewers for revising again the manuscript and the editor for their efforts. The changes in the manuscript are highlighted in green.

Reviewer 1

Comment: Regarding the modifications, I have just two remarks:

- Line 73: the authors mention that the “Aggregate data rate” is 4.4 Tbit/s. In fact, that is the “Aggregate net data rate”, or simply “net data rate”, which considers the overhead. Indeed, I would suggest the following: Provide in line 73 the aggregate data rate, or line rate, which is 4.8 Tbit/s, rather than the net data rate, due to the fact that no overhead is yet declared. Later on, when the overhead is specified, provide the aggregate net data rate, or simply net data rate, possibly together with the aggregate data rate. This should avoid possible confusions.
- Line 72: I would suggest to replace “data modulated” by “data encoded”.

Action taken: We thank the reviewer for the suggestions. In order to clarify that the 4.4 Tb/s is the aggregate data rate we now specify the error correction overhead. The specific sentence in the introduction now reads:

“Each channel contains data encoded using 20 GBd 64-quadrature amplitude modulation (QAM) resulting in an aggregate data rate of 4.4 Tb/s (assuming a 9% error correction overhead).”

Reviewer 3

Comment 1: The authors agree with the general sentiment that the work reported here should be judged primarily on the system results: “Our work represents a milestone in the field because we show that the net conversion efficiency of the comb source allows for modern coherent optical communications while keeping pump power level compatible with integrated silicon sources”. I am not qualified to judge the second claim, but the other two reviewers appear to be satisfied with this claim. The claim for modern coherent optical communications remains incorrect.

- Modern communications which would benefit from a comb source operate with higher spectral efficiencies, almost to the point where slightly exceeding the frequency stability of modern ITLAs would give a problem (a few GHz guard band). A guard band of 80 GHz is simply nearly 20 years out of date.

- Modern commercially available systems already operate at 600 Gbit/s per channel [<https://doi.org/10.1364/JOCN.9.000C12>] using 12 coded bits per symbol (PM-64QAM) and a symbol rate of 56 Gbaud. The net information spectral density of 600 Gbit/s of commercial systems is 10 bit/s/Hz (dual polarisation). This paper only presents results at 3.6 bit/s/Hz, approximately 1/3rd of commercially available systems. For a specialist journal where one may focus on optimisation of certain parameters, the results remain sufficient for publication. However, in terms of generating a “milestone in the field” I am afraid I disagree with this authors. In my opinion this paper does not represent a milestone in modern coherent optical communications. The device level measurement of beat note linewidth does not add or detract from the milestone nature of the systems result.

Response: We disagree with the reviewer statement that “the work reported here should be judged primarily on the system results” because it demerits the key novelty of the manuscript and would give more importance to e.g. the limitations given by the available equipment. Our work represents the first demonstration of a dark pulse comb in the context of coherent communications. These results indicate that the performance is compatible with advanced modulation formats (PM-64QAM), and that this can be attained while keeping a pump power level on the order of 100 mW. We believe this is a significant milestone when it comes to integrated comb sources as we can successfully perform high-order modulation while keeping the pump power level compatible with chip-scale lasers.

We agree that further optimization of the dark-pulse comb states towards 50 or 100 GHz line spacing enabling dense WDM with high spectral efficiency is an important future development. This aspect is extensively discussed in the main manuscript and the supplementary sections. However, whereas we have not achieved this experimentally, we believe that the results shown in this work are of sufficient value to show the potential of dark pulse combs and will encourage further investigation in order to reduce the line spacing.

Comment 2: The authors have now added an appropriate measurement to address this point. The beat note bandwidth of 30kHz is indeed lower than that which would be observed from some free running lasers, but is significantly higher than that which is observed on other comb sources. I disagree with the author’s conclusions from this data, but sufficient data has been presented for me to come to my own opinion so this is OK. However the true test of the suitability of the laser to the proposed application is a test of the system performance at a high spectral density, either directly or by emulation. Therefore I do not feel that the authors have done enough to address my concern to merit publication in this journal, although the new results make the manuscript even more suited to a specialist journal.

Response: We are glad to see that the reviewer is satisfied with the additional data provided. The level of stability of the line spacing of the comb is in pair with advanced semiconductor integrated mode-locked lasers and better than what can be obtained with free running lasers. Improved stabilization could be attained by locking the line spacing to an external radio-frequency reference (see e.g. ref. 43 in the main manuscript). However, state of the art narrow guard bands operate at 10 MHz (ref. 44), so we are far below this limit.

Comment 4: I welcome the authors’ acknowledgment of other works in the introduction, this is helpful to the reader. However, a source is either fully integrated and so only requires one fiber pigtail and fits inside a semiconductor laser package, or it is not fully integrated and requires multiple pigtails (for the device and for other components) and the eventual package would more resemble that of an EDFA. The author restating their claims that this has a few attractive characteristics does little to convince me that a fully integrated transmitter is enabled by this technology.

Response: We would still like to argue that the most fundamental roadblocks have been removed by the suggested solution. To further explain our reasoning we have slightly extended the related discussion paragraph.

Action taken: The following sentences have been added to the discussion section: “We envision that further device optimizations (i.e. larger resonators with optimal coupling and chromatic dispersion values) together with advances in integrated photonic circuits⁵⁰, will allow approaching a more complete integrated transmitter. By additionally restricting the system requirements to around 100 mW of pump power, the co-integration of a pump laser would become feasible. We also anticipate the expected channel symbol rates to keep increasing in the near future, making a 100 GHz spaced comb the final target. As described in this work, an optimized dark-pulse microresonator comb has the potential to empower such a system.”

Comment 5: I would agree with the authors that “the main feature in this paper is the possibility to attain high-OSNRs and stable line spacing while keeping a pump power level compatible with state of the art hybrid silicon lasers”. However, this type of excellent engineering development is more suited to a specialist journal. The authors acknowledge that they have made no attempt beyond this to influence thinking in terms of overall system design.

Response: As mentioned in the response to the first comment, in this manuscript we demonstrate the first usage of dark-pulse combs in microresonators for coherent optical communications. We appreciate the reviewer’s concern but disagree with the assessment of the long-term implications of the presented work. The combination of high OSNR and moderate pump power with the successful transmission experiment and the detailed discussion towards 50-100 GHz spaced dark pulse combs provides a clear path forward for using this technology in coherent optical communications. We believe this manuscript will trigger further investigation on dark pulse combs for coherent communications and we envision that research will be done to reduce the line spacing given the promising results shown in this work.

Comment: Most of the reviewer’s comments seem to have been addressed adequately to me, except; Comment 4: As the reviewer states “When using high order modulation...the stability performance of the comb lines is also important. More discussion on this performance should be covered”. The authors have provided a measurement but do not discuss or experimentally demonstrate what levels of frequency jitter and what types of jitter spectrum are required (their beat note linewidths do not look Lorentzian).

Response: As the minimum permissible guard band in a dense WDM system is set by a combination of all the surrounding equipment (including pulse shaping possibilities as well as both RF and optical filter bandwidths), we do not anticipate the comb line variations to come close to being the limiting factor in this regard.

Action taken: The beat note linewidth fits with a Gaussian, this is now mentioned in the text. We have also included a reference with state-of-the-art narrow guard bands of 10 MHz: “The beat note (see Fig. 1e) displays a clear peak >50dB above the noise floor (with a Gaussian fit FWHM < 30 kHz) indicating stable mode-locking operation beyond what is required in state-of-the-art dense WDM demonstrations⁴⁴.”